# Psychological Impact of Corona Lockdown in Germany: Changes in Need Satisfaction, Well-Being, Anxiety, and Depression

**DOI:** 10.3390/ijerph17239083

**Published:** 2020-12-05

**Authors:** Malte Schwinger, Maike Trautner, Henrike Kärchner, Nantje Otterpohl

**Affiliations:** 1Department of Psychology, University of Marburg, 35032 Marburg, Germany; maike.trautner@uni-marburg.de (M.T.); henrike.kaerchner@uni-marburg.de (H.K.); 2Department of Psychology, University of Giessen, 35394 Giessen, Germany; nantje.otterpohl@psychol.uni-giessen.de

**Keywords:** Sars-CoV-2, pandemic, lockdown, basic psychological needs, autonomy, well-being, depression, anxiety, mental health

## Abstract

All over the world; measures have been implemented to contain the novel Sars-CoV-2 virus since its outbreak in the beginning of 2020. These measures—among which social distancing and contact restrictions were most prominent—may have an overall effect on people’s psychological well-being. The present study seeks to examine whether lockdown measures affected people’s well-being; anxiety; depressive symptoms during the lockdown and whether these effects could be explained by reduced satisfaction of the basic psychological needs of autonomy and relatedness. *N* = 1086 participants of different ages and educational levels from all over Germany reported strong declines in autonomy and well-being; small declines in relatedness satisfaction; moderate increases in anxiety and depressive symptoms. These effects were stronger for people with moderate to bad subjective overall health. Latent change modeling revealed that, especially, decreases in autonomy satisfaction led to stronger decreases in well-being as well as stronger increases in anxiety and depressive symptoms; whereas decreases in relatedness had much weaker effects. Our results imply differential effects depending on individual preconditions; but also more generally that peoples’ need for autonomy was most strongly affected by the lockdown measures, which should be considered as important information in planning future lockdowns.

## 1. Introduction

As of mid-November 2020, more than 53 million people worldwide have been infected with the novel coronavirus Sars-CoV-2 and more than 1,305,000 people have died in connection with Covid-19 [1]. To stop the ongoing coronavirus pandemic, so-called lockdowns have been adopted in numerous countries. Although these lockdowns vary in the number and types of measures taken, they usually include certain types of contact restriction as well as the closure of many public meeting places, such as schools, public authorities, and shopping centers. Psychologists, doctors, and other health experts pointed out at an early stage that such interventions in individual lifestyles are likely to be accompanied by psychological consequences, such as increased anxiety or diminished well-being [2,3].

Such negative consequences are also to be feared from the perspective of prominent psychological theories. Self-determination theory states that people can go through life happily, productively, and motivated if three basic psychological needs are fulfilled, namely autonomy, competence, and relatedness [4]. On the other hand, if these needs are not met, it is to be expected that satisfaction with life and psychological well-being will decrease, while negative aspects, such as anxiety and depressive symptoms will increase [4,5]. To date, there is a growing number of studies dealing with the effects of isolation and quarantine with regard to COVID-19. For example, an Austrian study reported higher prevalence of mental health problems as the pandemic progresses [6]. Other studies revealed a high use of protective behavior [7] and a rather low level of leisure time sport activities [8] during the pandemic. A certain number of studies concluded that the societal conditions surrounding COVID-19 increase gender, social, and educational inequalities [9,10]. All of these findings are in line with reviews about psychosocial impacts of quarantine measures during previous pandemics that consistently showed negative effects on mental health outcomes [11]. Shortcomings of most available studies on mental health problems during COVID-19 are (a) that they did not investigate in how far respondents’ usual behavior has changed, according to the societal restrictions and (b) that they lacked a clear-cut theoretical rationale. In order to further contribute to this line of research, and using the example of the first coronavirus lockdown in Germany, we seek to investigate in the present study to what extent the restrictions in public life, which lasted for about three months from the end of March to begin of June 2020, (a) contributed to a reduction in psychological well-being and an increase in internalizing symptoms (anxiety and depressive symptoms) and whether (b) these changes can be explained by the reduced satisfaction of basic psychological needs for autonomy and relatedness. The results reported here are intended to enhance our understanding of the psychological processes underlying societal freedom restrictions as well as to provide guidance as to what political actors should pay particular attention to in the event of new restrictions. In view of the fact that, as of mid-November 2020, the number of new infections is skyrocketing in almost all European countries [1] and further lockdowns are on the way, we consider a solid analysis of the psychological consequences of such lockdowns to be particularly relevant.

### 1.1. Corona Lockdown in Germany

On 22 March 2020, the German federal and state governments agreed on a comprehensive “restriction of social contacts”. This included reducing physical proximity as much as possible, minimum distances between people in public spaces of at least 1.50 m, staying in public space only alone or with another person or in the circle of members of one’s own household. Moreover, groups of celebrating people—even in private—were expressly forbidden (however, it was made clear from the beginning that the inviolability of one’s own home would not be revoked). Sit-in areas of restaurants were closed. Personal hygiene service providers, such as hairdressers and beauty salons were closed, except medically necessary services. Generally, people had to comply with hygiene regulations and protection measures, such as wearing face masks. In addition to the jointly agreed measures, several federal states imposed even stricter exit restrictions, which made leaving one’s own home or entering public space fundamentally dependent on the existence of a “valid” reason. Beyond that, there were hardly any regional differences in the types and lengths of lockdown measures. In the second half of April, all federal states successively decided to make it compulsory to wear everyday masks on public transport and in shops.

In the period between the end of March and the beginning of June, the restrictions were extended approximately every two weeks. The citizens were always informed in advance of the next review date. Most of the restrictions imposed were gradually relaxed. From the beginning of May, for example, church services and prayer meetings were allowed to take place again, playgrounds reopened, and several cultural facilities, such as museums, were able to reopen under conditions of hygiene, access control, and avoidance of queues. Likewise, all shops were allowed to open under certain conditions and open-air recreational and leisure sports were permitted, subject to hygiene and disinfection measures and the maintenance of a minimum distance. The most important relaxation occurred at the beginning of June with the lifting of major contact restrictions. Nevertheless, as of mid-October 2020, public life in Germany is still far from its former normality. For example, distance rules and compulsory masks still apply in public places, and major events, such as music concerts, are still prohibited. Furthermore, stricter rules apply in places with more than 50 new infections per 100,000 inhabitants.

### 1.2. Satisfaction of Basic Psychological Needs

From the perspective of self-determination theory (SDT) formulated by Deci and Ryan [12], the above-described contact restrictions during coronavirus lockdown can be interpreted as a condition that reduces satisfaction of peoples’ basic needs for autonomy and relatedness [13]. Deci and Ryan formally identified the needs for autonomy, competence, and relatedness as basic psychological needs, arguing that support for and satisfaction of these needs accounts for a broad variety of phenomena across developmental periods, cultures, and personality differences. Basic psychological needs were broadly defined as critical resources underlying individuals’ natural inclination to move towards increasing self-organization, adjustment, and flourishing. Abundant research on SDT’s basic psychological needs (need for competence is not considered in this study) has shown that they indeed play a prominent role in development, adjustment, and wellness across cultures, with strong implications for motivational science, applied practices, and social policies [12,14,15]. Whereas past research has mainly included a measure of need satisfaction, recent investigations have also included measures of need frustration. The reason for doing so is that the absence of need satisfaction (i.e., need deprivation) cannot simply be equated with the presence of need frustration, as the latter implies a more active obstruction and undermining of psychological needs [13]. For example, if adult workers happen to learn a few new things on a given day, thereby experiencing little mastery, they may not necessarily feel like a failure. Additionally, although some workers may feel little connection with others, they may not necessarily feel lonely or alienated from others. Just as the absence of need satisfaction does not imply the presence of need frustration, more is needed for individuals to thrive than the absence of need-frustrating experiences; their needs also have to be nurtured and satisfied. SDT-based research has intensively studied key practices and correlates with need-supportive (i.e., autonomy support, structure, warmth) contexts [12]. Longitudinal studies spanning different time frames, from a few months to a decade, have revealed that need support predicts adjustment over time, as indexed by improved executive functioning, increased engagement, better emotional regulation, and higher achievement and well-being—see [13] for summary. In contrary cases, adolescents are prone to various types of psychopathology, such as internalizing problems [16].

### 1.3. The Present Research

In the present study, we seek to examine the psychological consequences of the coronavirus lockdown in Germany between the end of March and beginning of June 2020. Based on the widespread empirical evidence on SDT’s basic psychological needs, and the provision of need-relevant conditions, we propose that the contact restrictions during the coronavirus lockdown have deprived individuals’ needs for autonomy (due to the external conduct regulations) and relatedness (due to the external contact regulations). This kind of reduced basic need satisfaction, in turn, is supposed to result in a decrease in psychological well-being, as well as an increase in indicators of ill-being, specifically anxiety and depressive symptoms [17]. Moreover, we seek to examine whether older people, those with poor general health, and families with children were more affected by the lockdown.

## 2. Materials and Methods

### 2.1. Sample, Procedure, and Measures

The study followed the ethical guidelines of the authors’ universities, the professional ethics guidelines of the German Psychological Society (DGPs), the American Psychological Association’s Ethical Principles, and was in consultation with the ethics committee of the Department of Psychology at the University of Marburg (an ethical approval was not required for this study). Participation was voluntary and all participants were comprehensively informed about the information on study about aims, procedures, data protection and concept. All gave their electronically informed consent for inclusion before they participated in the study. The online survey did not contain any personally identifying details.

Data were collected via an online survey between 1st April and 28th May 2020 that was distributed via institutional mailing lists, social media, and private contacts. *N* = 1086 persons (74.8% women; age: *M* = 29.41, *SD* = 11.78, *Min* = 18, *Max* = 81) from all over Germany participated. In total, 15.8% reported having children. In total, 14.3% reported to have a moderate, bad or very bad overall subjective health status, the remaining reported to be in good or very good health. As their current occupation, 51.7% reported being a student, 3.8% studying and working simultaneously, 8.7% were research staff from different domains, 4.1% were employees in different, not further specified jobs, 4.7% were social workers and psychologists in different occupations, 1.7% were unemployed, 3.1% were doctors and other medical staff, 1.5% reported being creative artists or working in the cultural sector, 1.4% were retired, 0.6% worked in retail, 0.4% were freelancers or self-employed, and the rest came from various different, but mainly skilled occupations (lab assistants, teachers, craftsmen, technical assistants, engineers, project managers, journalists, veterinary assistants etc.).

First, participants reported demographic information on themselves and their household. Further, they were asked to report their general state of health on a five-point scale (“How would you generally describe your current state of health? very good/good/moderate/bad/very bad”). All further scales were administered twice in two columns next to each other, one asking for need satisfaction at “before the lockdown” (T1—i.e., around March), one for “today” (T2—i.e., between April and May, depending on the individual time of participation). Autonomy and relatedness satisfaction were assessed with the respective subscales from the German version of the Basic Psychological Need Satisfaction and Frustration Scale [11]. Both scales contained four items (e.g., autonomy: “I feel a sense of choice and freedom in the things I undertake.” relatedness: “I feel connected with the people who care for me, and for whom I care.”) that were answered on a scale from 1 (“do not agree at all”) to 5 (“agree completely”).

Well-being was assessed with four items (e.g., “I am satisfied with my present life”) of the German adaptation of the Temporal Satisfaction with Life Scale [18]. Participants stated their agreement on a scale from 1 (“not at all true”) to 4 (“very true”). To assess anxiety and depression levels, the trait subscales for depression-dysthymia and anxiety-worry of the State-Trait-Anxiety-Depression-Inventory (STADI) [19] were used. Each contained three items (e.g., dysthymia: “I feel sad”; worry: “I am worried something might happen.”) that were answered all on a scale from 1 (“seldom/never”) to 4 (“always”). All scales showed acceptable to good internal consistencies (see Table 1).

### 2.2. Statistical Analyses

The analysis of changes presupposes that the items measure the same underlying constructs at both measurement points. In addition to configural measurement invariance (measurement of a construct with the same items at different time points), metric measurement invariance (equation of all factor loadings over time) was checked first. If confirmed, it can be assumed that the latent constructs at both time points have the same meaning. Based on this, the extent of scalar measurement invariance was determined by the additional fixations of the corresponding intercepts. If scalar measurement invariance is proven, it can be assumed that there are no item-specific mean differences between time points.

After establishing measurement invariance for all constructs, scale mean differences over time were examined via t-tests and Cohen’s ds. Potential moderator effects for older people, persons with poor general health, and families with children were examined in 2 × 2 repeated measures ANOVAs. Relationships between the changes in the respective factors were examined via latent change models computed in Mplus 8 [20]. Latent change variables were specified for both need satisfaction and well-being (see Figure 1A for an example). Subsequently, these five variables were used to specify a structural equation model in which latent changes in autonomy and relatedness satisfaction predicted latent changes in life satisfaction, depression, and anxiety (see Figure 1B).

## 3. Results

Partial scalar measurement invariance was found for autonomy and relatedness satisfaction; full scalar measurement invariance was found for well-being, depression, and anxiety (detailed results can be obtained by the authors upon request). As displayed in Table 1, participants experienced significant changes in all variables over time (*t*s between |4.62| and |22.30|, *p*s < 0.001). Based on the effect sizes *d*, however, only the declines in autonomy satisfaction and well-being can be considered large, while all other effects were rather small to moderate. Moderator analyses revealed no significant effects for participants from different age groups nor for families with children (*F*s < 1). Participants of all age groups showed similar changes in need satisfaction (Figure 2), except for people aged 51 or older, who reported slightly larger, but still small declines in relatedness. Interestingly, life satisfaction decreased more strongly (with moderate to strong effect sizes) in both 18–25- and 51–81-year olds, but not so much in 26–50-year olds. A similar pattern emerged for anxiety and depressive symptoms, whereby the increases in both were moderate in these subsamples as opposed to small. Families with children living in the household experienced similar declines in need satisfaction as the overall sample (Figure 3), but strong declines in life satisfaction and medium to strong increases in depression and anxiety symptoms. Participants living with at least one child reported only small decreases in autonomy, but a slightly larger decrease in relatedness than the full sample. Their life satisfaction decreased more strongly, and anxiety and depressive symptoms increased more strongly (moderate effects). People who participated before or after mid-April showed similar decreases in autonomy satisfaction as the full sample; however, participation after mid-April resulted in slightly stronger decreases in relatedness, but smaller decreases in life satisfaction. Additionally, increases in anxiety and depressive symptoms were only small to moderate. In contrast to those rather descriptive differences, substantially worse changes could be identified for people with moderate to bad general health (*N* = 128) who showed a stronger decline in autonomy, relatedness, and well-being, as well as a stronger increase in anxiety and depressive symptoms (*F*s between 11.99 and 24.86, *p*s < 0.01; see Figure 4).

A latent change model with scalar (or partial scalar) measurement invariance for the respective scales fit the data well (*χ*^2^[569] = 1613.316; Comparative Fit Index (CFI) = 0.947; Standardized Root Mean Square Residual (SRMR) = 0.059). As expected, declines in autonomy satisfaction significantly predicted declines in well-being (*β* = 0.55, *p* < 0.001) as well as increases in depression (*β* = −0.44, *p* < 0.001) and anxiety (*β* = −0.47, *p* < 0.001). Likewise, declines in relatedness satisfaction predicted declines in well-being (*β* = 0.23, *p* < 0.001) as well as increases in depression (*β* = −0.31, *p* < 0.001) and anxiety (*β* = −0.13, *p* = 0.031). However, the effects of relatedness were considerably lower than those of autonomy satisfaction.

## 4. Discussion

With the present study, we sought to examine the psychological consequences of the three-month coronavirus lockdown in Germany between March and June 2020. Based on evidence from self-determination theory, we expected people to perceive lowered satisfaction of their basic psychological needs for autonomy and relatedness, which, in turn, was supposed to diminish subjective well-being and/or enhance anxiety and depression. Results were overall in line with our suppositions, indicating that autonomy, relatedness, and psychological well-being decreased over time, while symptoms of anxiety and depression were higher after a few weeks in lockdown. However, although all changes were statistically significant, strong effects were obtained only for perceived autonomy satisfaction and well-being. In contrast, moderate effects were observed for anxiety and depression and just small changes in relatedness.

Regarding the unexpectedly small effects for relatedness, we presume many people found alternative ways to interact and communicate with their peers via social media tools, such as WhatsApp, Instagram, or Facebook [21] This may have compensated for the restriction of personal contacts to at least some degree. This interpretation is supported by a survey conducted in July 2020 by the Federal Ministry of Health, in which the majority of German citizens (53%) stated that they had experienced no difficulties in complying with contact restrictions [16]. This may also explain why elderly people who are not as accustomed to using social media as younger people experienced a slightly larger decrease in relatedness. Regarding the moderate changes in anxious and depressive feelings, one might speculate that, for most of the people examined here, the weeks of lockdown may have been unpleasant, but not so emotionally stressful that ill-being has increased significantly. This interpretation is indirectly supported by the finding that 65% believe that the measures in March to combat the coronavirus pandemic were correct. Just about 20% each consider the measures to be too strict or think they should have been stricter [22]. Furthermore, people may have become used to the circumstances over time as they were able to observe falling infection numbers after being more strongly affected by the threat of the virus and resulting insecurities, which may also explain the stronger increases in anxiety and depressive symptoms.

It overall seems that many people have coped relatively well with the lockdown. However, we need to be very cautious in over-generalizing our results since we found that changes were much more pronounced in people with poor general health. This subgroup reported moderate to high influences on autonomy, well-being, anxiety and depression, thereby drawing a harsher picture of the situation. Future political decisions should therefore be tailored more closely to the particular sensitivities of different subgroups. Further to note, the sample was not representative and there are undoubtedly numerous individual aspects that our data cannot adequately reflect.

Latent change models revealed support for our expectation that changes in autonomy need satisfaction have implications for changes in well-being, whereas changes in relatedness were hardly related to the change in well-being, anxiety, and depressive symptoms. These findings highlight the importance of autonomy need satisfaction—i.e., the importance of being able to manage and control things in your own life [13]—leading us to conclude that future lockdowns or further measures at the societal level should be communicated in a way that takes particular account of the citizens’ sense of autonomy. Furthermore, there may be several more consequences of the Sars-CoV-2 virus and measures to contain is besides the lockdown itself (such as economic consequences impacting global markets, but also individuals due to unemployment and subsequent psychological consequences)—additionally, health-related concerns, such as fear of catching the virus oneself or suffering the consequences of limited abilities to exercise regularly, or the ability to cope with uncertainty regarding everyday life [23], which have not been accounted for in this study. Examining the precise mechanisms leading to need satisfaction or frustration and subsequent changes in well-being therefore seems necessary to shape more adaptive conditions during potential future lockdowns. These may have contributed to well-being and life satisfaction besides changes in psychological needs.

## 5. Conclusions

Overall, the present research implies the necessity to monitor psychological consequences of measures and implications to contain Sars-Cov-2 besides its physiological and psychological consequences of an infection with it. Regarding the most important limitation of this study, we acknowledge that, due to the quasi-longitudinal design participants’, estimations may have been biased by memory and contrast effects as participants had to recall their need satisfaction and well-being in hindsight. Consequently, declines in need satisfaction and well-being may have been enhanced artificially. Furthermore, the sample was not representative of the entire German population, limiting the generalizability of these findings. However, our findings do not largely differ from those studies relying on representative samples (e.g., [6]), so our data still provide insights into the psychological consequences of the 2020 coronavirus lockdown in Germany.

## Figures and Tables

**Figure 1 ijerph-17-09083-f001:**
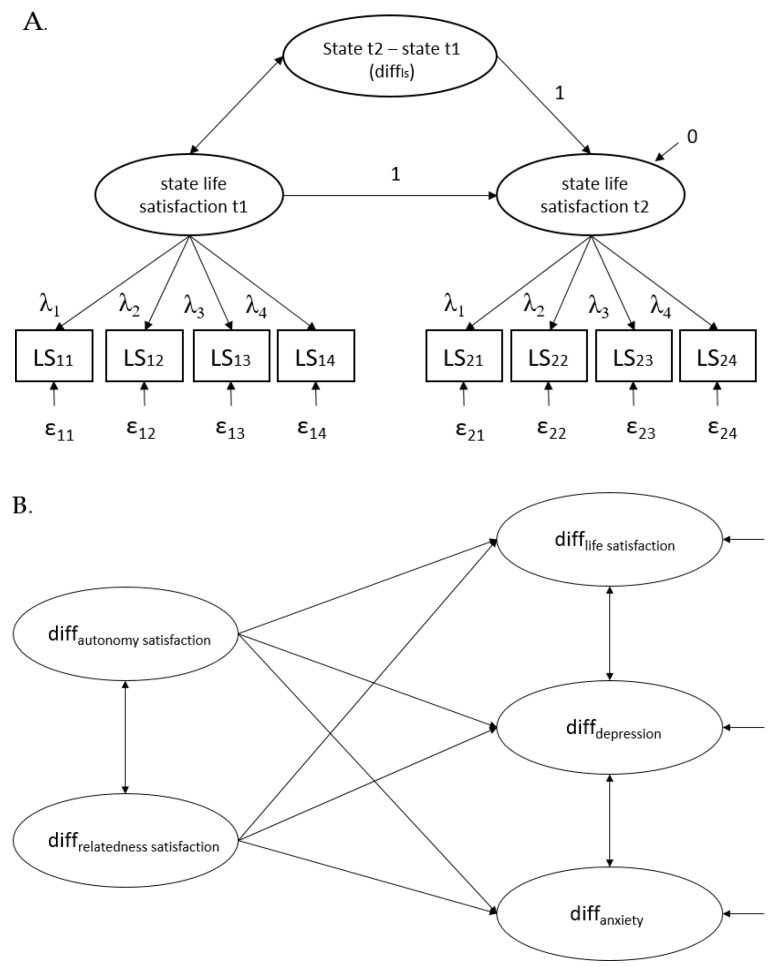
(**A**) Example of the Measurement Model Part of the Latent Change Model for Life Satisfaction (**B**) Structural Part of the Latent Change Model; standardized coefficients are reported. t1 = before the lockdown, t2 = today, LS_ij_ = life satisfaction observed indicator no. i at time point j (1 = before lockdown, 2 = now). ε_ij_ = residual of item no. i at time point j, λ_i_ = time-invariant factor loading of observed indicator i, regression coefficients of the latent difference variable (diff_ls_) and state life satisfaction at t1 were fixed at 1, the residual variance of state life satisfaction at t2 was fixed at 0.

**Figure 2 ijerph-17-09083-f002:**
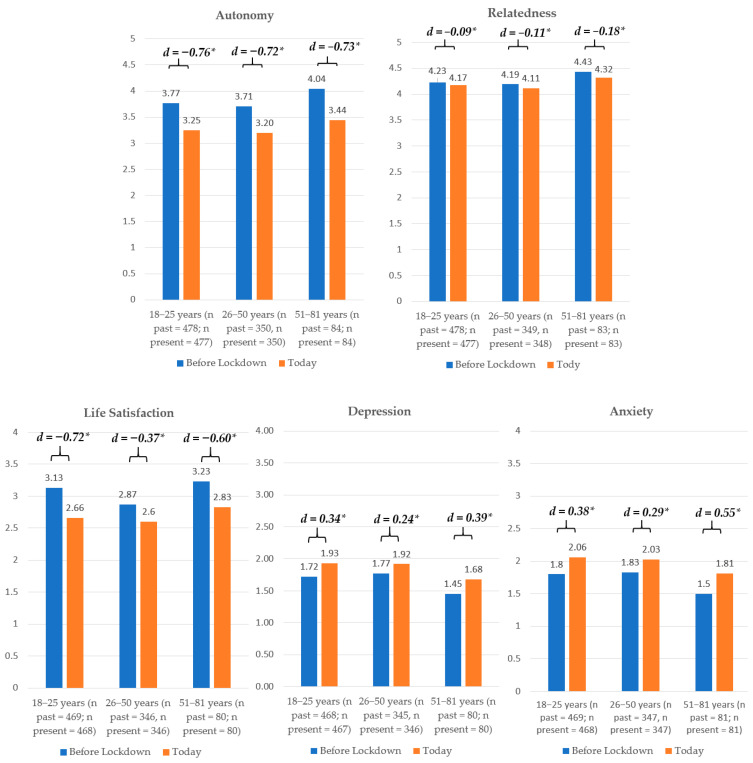
Differential Changes in Respective Age Groups. *d* = Cohen’s *d* (effect size), * *p* < 0.05.

**Figure 3 ijerph-17-09083-f003:**
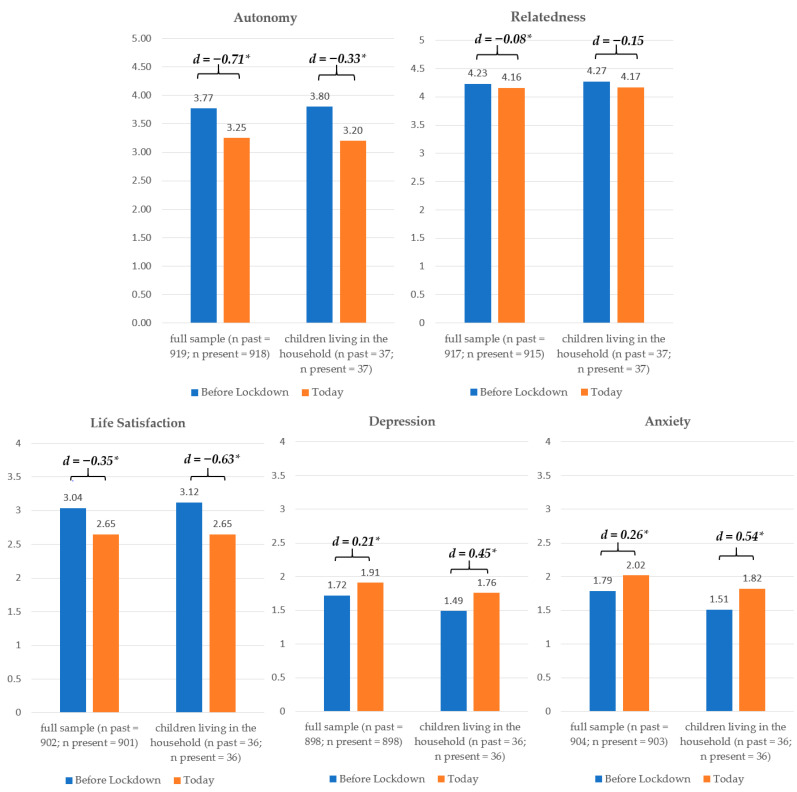
Differential Changes in the Full Sample and for People with Children Living in Their Household. *d* = Cohen’s *d* (effect size), * *p* < 0.05.

**Figure 4 ijerph-17-09083-f004:**
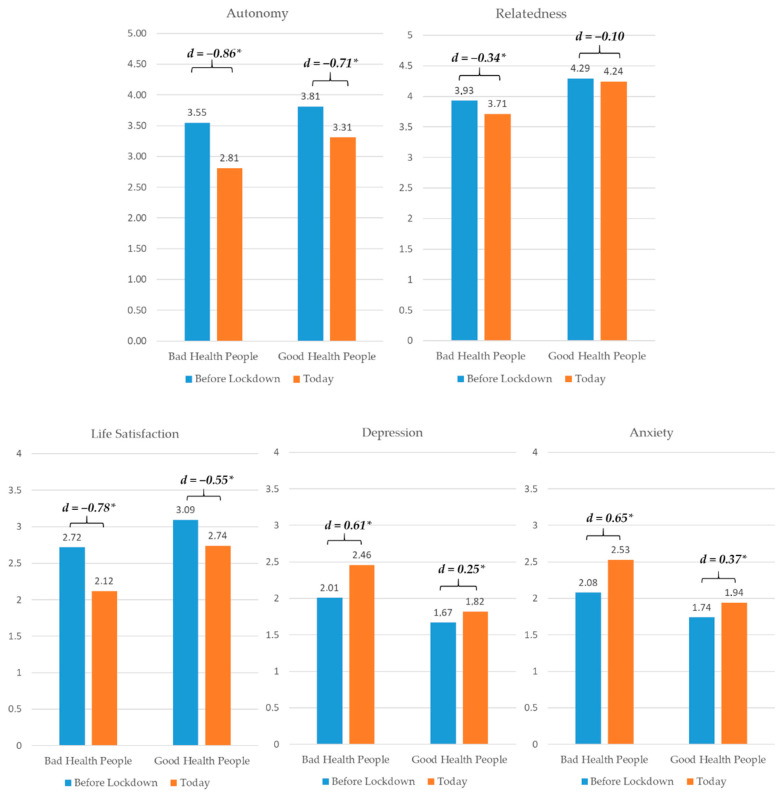
Differential Changes for People with Bad vs. Good General Health. *d* = Cohen’s *d* (effect size), * *p* < 0.05.

**Table 1 ijerph-17-09083-t001:** Scale Means, Standard Deviations, Reliabilities, and Cohen’s d’s.

Scale	*T1*		*T2*		*Mean Differences*	
	*M (SD)*	*α*	*M (SD)*	*α*	*t (df)* _T1–T2_	*d* _T1–T2_
(1) Autonomy	3.78 (0.71)	0.76	3.25 (0.74)	0.73	−22.30 * (917)	−0.71
(2) Relatedness	4.24 (0.67)	0.82	4.16 (0.74)	0.80	−4.26 * (914)	−0.15
(3) Well-Being	3.04 (0.65)	0.89	2.65 (0.71)	0.83	−17.33 * (900)	−0.59
(4) Anxiety	1.79 (0.64)	0.74	2.02 (0.70)	0.78	12.47 * (902)	0.40
(5) Depression	1.71 (0.58)	0.80	1.91 (0.69)	0.83	8.94 * (896)	0.32

Note. *T1* = Before Lockdown, *T2* = Today. M = Scale Mean, SD = Standard Deviation, *α* = Cronbachs alpha, *t* = value of t-distribution, df = degrees of freedom, d = Cohen’s d (effect size measure). * *p* < 0.05.

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
