# Peer review of "Psychological Impact of Corona Lockdown in Germany: Changes in Need Satisfaction, Well-Being, Anxiety, and Depression"

_ijerph, 2020, doi:10.3390/ijerph17239083_

Round 1

Reviewer 1 Report

The study is based on a clearly defined research question. The study population, the main variables and the type of relationship expected between them are adequately mentioned.
The study sample was not probabilistic and no data is provided to evaluate its similarity to the base population, nor the measures adopted to minimize possible selection bias, all of which makes it difficult to compare between age groups and to generalize the results. Although we understand that they carried out the study in the best possible way given the circumstances (confinement of the population), we consider that these aspects should be considered in the discussion of the results for a better assessment of them.
In general terms, the measurement of the study variables was done in an adequate manner. Their role in the study is clearly stated, and they are defined in a conceptual and operational way. The instruments for measuring the variables have known validity and reliability, although it would have been more appropriate to offer data on the reliability of the measurements in the sample examined, including their factorization, given the use of the dimensions of these variables in the study. In addition, authors should consider the possibility of recall bias on the part of respondents, since they were required to recall their experience prior to confinement.
Statistical analyses are adequate, but have not taken into account other confounding factors or variables that affect well-being linked to the satisfaction of basic psychological needs for autonomy and relationship; for example, the people with whom the respondent lives.
Considering all of the above, it does not seem that the study has sufficient internal validity.
The results are well described, precise and useful for validating the theory of self-determination, but their practical utility is questionable because the sampling system does not facilitate the generalization of these to the general population, although it does to possible sectors of the population.

Reviewer 2 Report

Important, well-designed, and clearly presented.  A few minor word order changes and edits needed to conform to standard English.

Reviewer 3 Report

Thank you for submitting this paper. It was very well written and the study well constructed. I realise you were limited in the length of report you could write but I was interested that you didn't cite any of the existing literature on isolation and quarantine with regard both to COVID-19 and previous epidemics and pandemics. There is an extensive literature on this topic. For clarity and to better understand the psychological impact of lockdown it would have been useful to know if people knew from the outset how long lockdown would last; or if there were major regional differences in the length/type of lockdown. 

There is only one minor point that needs clarifying: was the practice of celebrating in groups (?Parties?) 'unacceptable' or expressly forbidden? It's not clear and would have quite large implications. 

Reviewer 4 Report

Dear Authors,

Hope you stay safe and healthy. Thank you for the excellent paper, I really enjoyed reading it. The topic is important and relevant to the current situation of pandemic and lockdown in many countries worldwide.

The study is well-hypothesised with respect to the validity of the scales used. Statistical analysis is excellent taking into account measurement invariance in different time points and moderating effects of self-rated health. The results are well presented and discussed. The conclusions support the main findings.

I have several comments and suggestions which might add more value to the paper.

Please add a description of self-rated health to the methods section.

If I understand properly, linear regression was used to predict well-being, anxiety and depression. If yes, please add this information to the statistical analysis section.

Please reconsider providing tables of multifactorial ANOVA and linear regression to represent changes of the study variables during the lockdown and predicting effects of the decline in autonomy on depression and anxiety level. I believe this would add strength to the paper.

There are more factors affecting the psychological well-being of populations during the lockdown together with limited contacts and threats of unemployment. For example, the limited possibility for physical activity especially outside and prolonged duration of sedentary time communicating, learning and working online may affect the mental health of people of different ages. Moreover, danger and fear of being infected by coronavirus may add a negative effect on elderly people psychological well-being. I suggest adding a more detailed discussion taking into account the factors listed above.

Please reconsider adding limitation of the non-country representative sample.

Round 2

Reviewer 1 Report

I appreciate your response and consideration of my comments. I sincerely believe that your work is now much more precise and clear, thus allowing other researchers to extract good guidelines for possible future studies and to accurately assess the scope of the factors analyzed. I reiterate my thanks for considering my comments in such detail.

Reviewer 4 Report

Dear Authors,

Thank you for the improved version of your manuscript. Wish you all the best in your further research.